# Ubuntu Is a Critical Component in the Fight against Human Immunodeficiency Virus and Tuberculosis Stigma: Nursing Students’ Perceptions

**DOI:** 10.3390/ijerph21020229

**Published:** 2024-02-15

**Authors:** Melitah Molatelo Rasweswe, Nancy Mamoeng Kgatla, Irene Thifhelimbilu Ramavhoya, Fhumulani Mavis Mulaudzi

**Affiliations:** 1Department of Nursing Science, University of Limpopo, Private Bag x1106, Sovenga 0727, South Africa; nancy.kgatla@ul.ac.za (N.M.K.); irene.ramavhoya@ul.ac.za (I.T.R.); 2Department of Nursing Science, University of Pretoria, Private Bag x323, Arcadia 0007, South Africa; mavis.mulaudzi@up.ac.za

**Keywords:** HIV, nursing students, perceptions, TB, stigma, Ubuntu

## Abstract

Stigma is one of the documented barriers to achieving universal access to human immunodeficiency virus (HIV) and tuberculosis (TB) prevention, treatment, care, and support programs. The lack of African social theories to emphasize these issues may be the cause of the continent’s failure to reduce stigma. We can use Ubuntu, an African philosophy that emphasizes sociability and ethics, to deepen our understanding of how to reduce HIV- and TB-related stigma in South Africa. In many African regions, Ubuntu values and principles were found to assist in reducing problems related to HIV. Ubuntu is a comprehensive phrase used by Africans to signify the characteristics that incorporate the core human virtues of compassion and humanity. We explored the perceptions of nursing students regarding the use of Ubuntu in the fight against HIV and TB stigma. Qualitative participatory research was used to engage all first- and second-level nursing students enrolled for the 2023 academic year at the selected university in South Africa. Purposive sampling was used to recruit the participants. An interactive workshop was used to gather data. The data from the recordings and flip charts were analysed together. Throughout the study, trustworthiness and ethical principles were upheld. Two primary themes emerged as expressions of Ubuntu in relation to a reduction in HIV and TB stigma and recognition of Ubuntu as a tool to combat stigma associated with HIV and TB. This confirms that within Ubuntu, there are elements that might be applied to reduce stigma attached to HIV and TB.

## 1. Introduction

Human immunodeficiency virus (HIV) infection and acquired immune deficiency syndrome (AIDS) have been present in South Africa for more than 40 years since their initial identification [1]. Since then, there has been a significant increase in the availability of life-saving HIV therapy, which is reducing the number of new infections [2]; however, an opportunistic disease known as tuberculosis (TB), which is caused by the *Mycobacterium tuberculosis* complex, is affecting more people living with HIV (PLHIV) [3]. The co-existence of these diseases poses unique challenges due to their close link, which has serious negative effects. In South Africa, the co-dependency of these diseases and the difficulties in implementing successful control measures has been previously examined [2]. South Africa has been battling with this co-epidemic of HIV and TB for decades and contributes to 50% of these infections within the African countries [2,4]. Both HIV and TB are highly stigmatized diseases that have been recently conceptualized in terms of healthcare and social stigma. Numerous individuals have confirmed being stigmatized by Health Care Providers (HCPs) for having HIV and TB. This stigma comes in many forms, such as overshadowing diagnosis and treatment, whereby HCPs link patients physical complaints to HIV and TB [5]. Some are unkind and display ineffective actions, which turn patients away from medical treatments, and threatening behaviour [6]. Many times, stigmatizing attitudes about HIV and TB by HCPs lead to patients’ own emotions of guilt and self-stigma. Therefore, because of pervasive stigma among HCPs, PLHIV and TB may decide not to seek HIV/TB support. Ignorance of the issue of stigma among HCPs can result in subpar treatment and worse health outcomes for PLHIV and TB. A study that unpacked the dynamics of double stigma of HIV and TB stressed the need for an integrated approach in the fight against HIV and TB stigma in both healthcare and social dynamics [5]. The same authors’ study results demonstrated that the perceived link between HIV and TB among HCPs is significantly associated with stigmatization [5].

The stigma by HCPs has been found to be particularly harmful, resulting in poor health outcomes for people with HIV and TB. Another study discovered that HCPs, including nurses, and nursing students compromise patients’ confidentiality by gossiping about their HIV status [7]. Some nurses and medical doctors reported that they had personally stigmatised and discriminated against PLHIV [8]. This results in HCPs transferring the views of HIV stigma to TB [9]. Therefore, the persistent stigmatization of HIV is altering the attitudes towards HIV and TB screening and treatment [9] and contributing to nurses and nursing students delaying and avoiding taking care of those who need HIV and TB care [10]. It also restricts people’s ability to seek and accept HIV and TB preventative treatments, participate in care, and stick with treatment [11]. Additionally, it will be challenging to meet UNAIDS’ proposed HIV reduction 95-95-95 targets, which call for 95% of PLHIV to know their status, 95% of those diagnosed to be on antiretroviral therapy, and 95% of those receiving treatment to have a suppressed viral load by 2025. The other fact is that South Africa’s TB death toll (both HIV-positive and HIV-negative cases) did not decrease linearly between 1990 and 2020, which raises concerns because the country has to cut the total number of TB deaths by 90% by 2030 [2]. The same authors contend that a more aggressive control strategy is desperately required to meet the 2030 target if TB cases are to drop by 90% and HIV by 95-95-95 [2]. To meet the above-cited ambitious and necessary targets, there is a need for continued and persistent efforts that are valued and accepted by the community. It is, therefore, critical to make sure that the stigmas related to HIV and TB are addressed concurrently among nursing students [12]. Nurses are regarded as the backbone of the health care sector and spend a lot of time with the patients [13]. They are also role models and mentors for nursing students. If qualified nurses continue to stigmatise PLHIV and TB, it will affect the nursing students’ personal and professional behaviour towards PLHIV and TB. Nursing students are the next generation of nursing profession and therefore need to be trained positively, especially on Ubuntu. As these students graduate and join the workforce, they will act as agents of change, lowering stigma and enhancing patient care using Ubuntu. Therefore, it is important to handle stigma related to HIV and TB early in their career, rather than later in their practice, because they are becoming more aware of role models and creating their own professional identities at that time.

The role of both healthcare and social dynamics in combating HIV and TB has been acknowledged by the government of South Africa [2]. However, the question of how to change the stigma and discrimination about HIV/AIDS and TB infections has caused concern. A significant number of individuals and states share the global belief that effective HIV and TB stigma and discrimination requires local context, such as Ubuntu, which incorporates human rights and societal norms [14]. Hence, many initiatives that recognise the local ways of knowledge and societal norms have been launched to tackle the negative perceptions and obstacles that communities encounter while seeking treatment for both HIV and TB. Research has shown that South African communities are guided by a variety of laws, principles, and ideologies, including Ubuntu [15]. Ubuntu is an African motto that emphasizes the importance of showing respect and compassion towards others [16]. It is the central element of African culture and influences every aspect of people’s lives [17]. In the African tradition, Ubuntu is rooted in the community, emphasizing respect, care, kindness, warmth, understanding, sharing, humanness, reaching out, and feeling connected, which is more similar to nursing care [18]. The concept of Ubuntu is used in most African societies, though it varies depending on the societal language [19]. In South African societies, the concept of Ubuntu is well known in all the eleven official languages. It is viewed as a factual description of societal ethics because it is based on philosophical values and principles that consider others as authentic human beings [20]. Ubuntu is taught mostly through actions in the community; therefore, the nursing students learn Ubuntu daily through social interaction within their community.

The adoption of Ubuntu values and principles in the past has provided a solid foundation for nursing care and healthcare ethics [17,21,22]. Literature provided by Albertina Sisulu and Ubuntu pointed out that Ubuntu encompasses values such as love, compassion, and peace that are significant for nursing and caring [23]. In another study, nursing students revealed that Ubuntu and nursing care are intertwined because they both centralise humanness, which strengthens the practice of caring [18]. Hence, many scholars are characterising Ubuntu as a philosophy to adopt in the fight against diseases that are easily stigmatized like HIV and TB. For example, Ubuntu was applied to improve nursing care in a hospital that was overwhelmed with HIV-positive patients [24]. The application of Ubuntu values and principles can assist in caring for people with HIV/AIDS and other diseases such as TB [25]. Some authors urge people to integrate Ubuntu philosophical values into HIV and AIDS prevention and management [26]. It is evident from the above argument that Ubuntu and nursing are interdependent; hence, it is important for African nurses, including nursing students, to use Ubuntu as a tool to eliminate stigma associated with HIV and TB. The authors of this paper argue that initiatives that are not integrating Ubuntu in reducing stigma associated with HIV and TB may not be effective or sustainable in South Africa. Nonetheless, South African HIV and TB stigma resiliency is evident through destigmatization initiatives, creative prevention interventions, and noteworthy contributions to international research [2]. However, the measures exclude the application of Ubuntu, which is believed to carry ethical weight and respect to reduce negative attitudes and position the worth of others. In order to close this gap, our study sought the perceptions of selected university nursing students regarding Ubuntu as a critical component in the fight against HIV and TB stigma. Involving nursing students in the knowledge of Ubuntu as a vital component in the fight against HIV and TB stigma gives them a valuable perspective on how to address stigma and create long-lasting programmes. Furthermore, the knowledge that nursing students contribute to this problem could instil a sense of ownership and apply this knowledge successfully in subsequent clinical practice.

## 2. Materials and Methods

### 2.1. Research Design and Setting

We used qualitative participatory research in order to engage the nursing students to use Ubuntu in fighting stigma against HIV and TB. The study setting was a university in Limpopo province, South Africa. The university has various faculties, one of which is the Faculty of Health Sciences. The Faculty of Health Sciences offers numerous health sciences programmes, including undergraduate and postgraduate nursing. The undergraduate nursing programme offered at the selected university is called R174, and it is regulated by the South African Nursing Council. It is a four-year degree designed to offer both theory and practice modules. The nursing science department admits students from all nine provinces in South Africa and neighbouring African countries, representing a diverse spectrum of cultural, racial, and ethnic backgrounds. The students from nine provinces of South Africa speak eleven different languages. Nursing students are placed for clinical practice from first through to fourth level at the university’s two teaching referral hospitals, as well as other district hospitals and primary health care clinics in the surrounding areas. The placement is on a rotational basis, depending on the objectives to be achieved. These clinical practice facilities cater for different healthcare services, including HIV/AIDS and TB, whereby nursing students may offer health care services to PLHIV and TB.

### 2.2. Participants and Sampling

We engaged all the first- and second-level nursing students enrolled for the 2023 academic year at the selected university in South Africa. After obtaining permission to conduct the study from the departmental head, the participants were purposively recruited in their respective classes through word of mouth. During data collection in December 2022, all the nursing students in the first and second level were of African origin, from Sepedi, Tswana, Ndebele, Tsonga, Tshivenda, Sesotho, Zulu, Swati, Xhosa, and Afrikaans (mixed race, known as coloureds). All the participating nursing students were eighteen years and older, and before attending a workshop they all signed written informed consent. There were 160 nursing students in attendance and they were divided into 16 working groups. Each group was composed of ten individuals and one person was nominated as a group leader.

### 2.3. Procedures

Data were collected through a workshop in order to explore, record, and interpret nursing students’ perceptions regarding the use of Ubuntu in fighting stigma against HIV and TB. A workshop is an environment where many people are engaged in a discussion to acquire knowledge, solve problems creatively, or innovate in response to specialized issues [27]. Three interactive workshops were held daily for four days in one setting. The workshops were conducted in the English language since it was a medium of instruction for the nursing students. The purpose was to sum up HIV/AIDS, TB, and Ubuntu to the participating nursing students at the beginning of the workshop. Although HIV/AIDS and TB were formally taught previously in class, the first three authors gave an overview of both concepts together with Ubuntu, as it was anticipated that Ubuntu was informally introduced in their irrespective communities. The authors also role-played some information on both HIV/AIDS, TB, and Ubuntu to represent a stigma position or voice related to HIV/AIDS and TB in order to trigger critical reasoning for the Ubuntu adoption. Daily, in their groups, nursing students were provided with activities to complete and present to the entire attendees. The prompt questions to stimulate group discussion regarding the use of Ubuntu as a critical component to fight against stigma associated with HIV and TB were either: (1). *What are the perceptions of nursing students regarding Ubuntu as a critical component in the fight against HIV/AIDS and TB stigma? or* (2). *How does Ubuntu influence fight HIV and TB related stigma?* All the groups’ discussions were audio-taped with the participants’ permission. Each group presented their responses to everyone, and when needed, probing questions were asked to elucidate the data. All the points were captured both audibly through recordings and visually on the flip charts.

### 2.4. Data Analysis

We transcribed all the recordings verbatim and added the data from the flip charts. The transcripts were read several times to familiarise the students with the content. Once the transcripts were understood, we created different worksheets for coding using an Excel program. Thereafter, all the authors were involved in identifying the commonly used phrases or words that were central to the group discussions separately, allowing generation of initial codes for data extracts. One author (M.M.R.) grouped the codes on a worksheet until a codebook with similar codes was developed. We held weekly meetings to discuss patterns of meanings and how those meanings related to the nursing students’ perceptions regarding Ubuntu as a critical component in the fight against HIV and TB stigma. Through the research team discussions, codes and subcodes were categorized and then reviewed for further similarities in order to form themes, making sure that the data made sense and that the themes were distinct and easy to understand. After conducting critical analyses of data sets and holding face-to-face weekly discussions, the team came to a broader consensus on the derived final themes and used the nursing students’ quotes to explain those themes. The transcripts and codes were shared with an independent coder to confirm the emerged categories and themes.

### 2.5. Trustworthiness

The research team adopted standards of trustworthiness according to Lincoln and Guba [28]. Credibility was proven by ongoing, interactive engagement with the participants and data. To increase the credibility of the interpreted data, the research team held weekly meetings to analyse the data. Dependability was established through coding, sub-categorizing, and categorizing. To ensure conformity, all the team members were engaged in the analysis and coding process.

## 3. Results

### 3.1. Perceptions of Nursing Students Regarding the Use of Ubuntu in the Fight against HIV and TB Stigma

Two main themes emerged from the analysis, as presented below.

#### 3.1.1. Expression of Ubuntu in Relation to Reduction in HIV and TB Stigma

Local language was considered as a driver for the implementation of Ubuntu to reduce stigma associated with HIV and TB. The nursing students indicated that it is important to know Ubuntu in ones own language before using it to reduce HIV and TB stigma. The participants explained in their own words the significance of using Ubuntu in their own language. They indicated that one’s own language positions the meaning and understanding of why Ubuntu should be used to lessen the stigma associated with HIV and TB. Group 8 participants indicated that “For a meaningful understanding and application of Ubuntu in fighting HIV and TB stigma, it is very easy when own language is used instead of Ubuntu”.

Group 13 verbalised that “Even though most of us are aware of what Ubuntu is, it has greater impact when expressed in one’s own tongue, which encourages application”.

In South Africa, depending on the community’s language, Ubuntu is also known as “Botho” in Sepedi, Tswana, and Sesotho, “Vumunhu” in Xitsonga, and “Vhuthu” in Venda. However, in the current practice, the “Nguni” word (Ubuntu), which covers Zulu, Ndebele, and Xhosa, is dominantly utilised. Based on the quotes, it was evident that a variety of South African languages and cultures share a common lexicon of Ubuntu knowledge. Additionally, instead of using Ubuntu, other participants decided to communicate in their mother’s tongue in order to show a bigger influence of it to combat HIV and TB stigma.

The nursing students also think that gestures and actions support the expression of Ubuntu to lessen stigma associated with TB and HIV. To reduce HIV and TB stigma, the nursing students spoke of Ubuntu expressions as being intentional and genuine to offer holistic services that accept PLHIV and TB as human beings. Gestures and actions were related to non-verbal indicators that are made when with PLHIV and TB. Group 2 said “When offering HIV and TB services, show them that you care about them. Don’t only give them medicine; instead, inquire about their overall health, including their psychological and spiritual well-being”.

Other nursing students emphasized that greetings, talking, and smiling to PLHIV and TB when providing services acknowledges their worth and they will not feel stigmatised. Group 1 highlighted that “We believe that to reduce HIV and TB stigma through Ubuntu/Botho HCPs can just greet the clients, you know to greet is Botho”

Group 15 said “It is Ubuntu to smile at individuals receiving HIV and TB services, and it can help lessen the stigma associated with these treatments”

The aforementioned highlights how Ubuntu can be conveyed in various ways and local languages to lessen stigma associated with HIV and TB. Ubuntu can thus be positioned or situated in different forms to reduce HIV and TB stigma.

#### 3.1.2. Recognition of Ubuntu as a Tool to Combat Stigma Associated with HIV and TB

The study findings demonstrate that the Ubuntu concept is extensively accepted as a weapon to fight stigma associated with TB and HIV. The nursing students recognised and acknowledged the importance of ethics, dignity, and respect in relation to fighting stigma associated with HIV and TB. The participants said Ubuntu upholds the same ideals and ideas, which are essential for removing the stigma associated with HIV and TB. Group 11 states that “Ubuntu teaches people to be ethically in whatever they do, therefore it is appropriate for use in reducing HIV and TB”. While group 16 uttered that “Ubuntu is moral; it carries ethics, it establishes standards for human behaviour and rules, particularly for vulnerable populations like those affected by HIV and TB”.

This finding demonstrates that, in the fight against the stigma attached to HIV and TB, ethics within the Ubuntu idea should be acknowledged and utilised. The findings also show that dignity is the fundamental component of Ubuntu for avoiding HIV and TB stigma. Dignity was related to being kind to PLHIV and TB, handling them well, and letting them talk openly about their conditions. Group 16 believed that “Ubuntu is well positioned to prevent HIV and TB stigma, because it promotes each person’s worth and supports their dignity”.

The same sentiments were shared by group 8, who also believed that dignity had an impact on combating stigma associated with HIV and TB. They said, “To us, preventing HIV and TB-related stigma, means providing dignified services, like treating clients well, with dignity, without discrimination, like they accessing the services without being labelled”.

In their discussion about preventing HIV and TB stigma, the nursing students made strong references to Ubuntu and dignity. They suggested broadening the application of Ubuntu to elevate humanity in PLHIV and TB to prevent stigma that is affecting the prevention and treatment of both diseases. Group 7 stated that “Ubuntu genuinely cares about people’s dignity and worth of individuals, thus it advocates for dignity and dignity liberates an individual from the oppressive influences and negative pressure”.

Group 9 felt that “Ubuntu dignifies all the people, it cares about people’s dignity, which frees them from negative mindsets and oppressive beliefs. Therefore, using Ubuntu to lessen the stigma attached to HIV and TB can be simple”.

In many instances, participants were exhibiting dignity and respect at the same time. They emphasised that dignity upholds respect for human uniqueness, explaining why an individual is who they are because of other people.

Group 4 stated that “Ubuntu is characterized by code of conduct like dignity and mutual respect. Since dignity comes from respect, treating PLHIV and TB with dignity also respects them as human beings, and the community as a whole will recognize them as valuable as any other individual and stop stigmatising their conditions”.

Another group also felt that stigma related to HIV and TB might be countered by treating people with respect and dignity. Group 12 participants said, “Ubuntu advocates for decency, therefore if applied in caring for PLHIV and TB, they will feel respected and not stigmatised”.

Based on the evidence from our data, it is clear that Ubuntu values ethics, dignity, and respect, all of which have a good impact on people’s behaviour when interacting with others. Ubuntu should therefore be accepted and acknowledged for its ability to alter people’s attitudes to TB and HIV.

## 4. Discussion

Our research lays the groundwork for utilizing Ubuntu to combat the stigma associated with TB and HIV. The themes that emerged during analysis included expression of Ubuntu in relation to a reduction in HIV and TB stigma and recognition of Ubuntu as a tool to combat stigma associated with HIV and TB. While there is limited literature on the application of Ubuntu in the prevention of stigma related to HIV and TB in South Africa, our findings are in line with earlier studies examining the values and principles of Ubuntu when addressing HIV-related concerns generally [24,25,26,29].

Additionally, the nursing students believe that referring to Ubuntu in the local language enhances its significance. They emphasised that it is very simple to comprehend and apply Ubuntu in a meaningful way to combat HIV and TB stigma when one uses one’s native tongue. Thus, before putting Ubuntu into practice, one must comprehend what it means in their own language. Most African civilizations share the idea of Ubuntu, though the name varies depending on the language [19]. Therefore, Ubuntu is a language that negotiates identity that is not dependent on race [30]; it can be conveyed in any language and yet has a significant impact on the user, despite linguistic variations. This demonstrates how well local language influences people to comprehend phenomena [31].

The terms “Botho”, “Vumunhu”, and “Vhuthu” originated from the narratives of nursing students in an effort to highlight the importance of local language when utilizing Ubuntu to prevent HIV and TB stigma. A study that exported ancient African wisdom into the global world argued that it is hard to pin down the African understanding of Ubuntu in one language because it derives its knowledge from the real world in which it grows [32]. This means that Ubuntu, in all other languages, appears to deliver special results in terms of battling the stigma attached to HIV and TB. Therefore, to have a positive influence on reducing stigma related to HIV and TB, Ubuntu must be named in the various local languages spoken in South Africa.

The participants also emphasized using gestures and actions to express Ubuntu. This aligns with the concept discussed in comprehending Ubuntu in action [33]. The author emphasizes the significance of exercising Ubuntu with intentions and sincere gestures and deeds. It is also recognized that gestures can be used to convey intentions [34], so that there is an act of owning up to the consequences of own actions in preventing stigma associated with HIV and TB. Moreover, gestures and actions were found to spread positive or negative attitudes about HIV and TB.

The concept of Ubuntu was also recognised by the nursing students as a tool to combat stigma associated with HIV and TB because in South Africa, Ubuntu is regarded as cultural legacy rooted in ethics, respect, and dignity. This finding concurs with the description of Botho/Ubuntu as the heart of African ethics [35]. According to the previous author description, Ubuntu is an ancient concept that is deeply driven and facilitated by African ethics and practices. African ethics are defined as a system of principles that are distinctly linked to respect and dignity [36]. It is also significant to highlight that African ethics are rules that proceed through common moral position to describe and regulate the behaviour [37]. In addition to being descriptive and regulative, Ubuntu was also argued as being a normative ethical statement guiding how we should treat each other and how to become human [20]. In addition, ref. [18] contends that Ubuntu promotes an atmosphere for practicing ethics, since it considers individuals as whole persons. Thus, it urges moral judgment and behaviour, which directs positive actions towards others, because Ubuntu represents firmly held African beliefs that a person’s identity is shaped by the relationships with other people [38].

According to the nursing students, the application of rules, norms, and morals embedded in Ubuntu will help to control the stigmatization and discrimination toward illnesses like HIV and TB because the affected will be treated with dignity. The authors that used the African community idea of Ubuntu to challenge stigmatisation of HIV and AIDS in a Zimbabwean Christian Hospital in the early years of discovering HIV revealed that Ubuntu upholds the human dignity and respect of other individuals [16]. It further showed that when PLHIV’s dignity is undermined and taken away, they feel disrespected and discriminated. Dignity is regarded as a fundamental aspect of human existence, which suggests that all people are created equal [39]. In relationships with others, dignity enables one to feel valued and significant, especially in times of threats like HIV and TB [40].

Therefore, putting Ubuntu ethics to use, one may prevent the stigmatization of HIV and TB and the hatred that fuels it. Additionally, treating PLHIV with dignity and respect will accept them as fellow human beings, especially in South Africa, where stigma and discriminatory attitudes are pervasive [41]. As a result, TB and PLHIV will be able to live dignified lives and experience the humanity that Ubuntu promises. The authors of this paper believe that Ubuntu ethics, dignity, and respect, combined with professional values, will motivate healthcare providers to refrain from stigmatizing PLHIV and TB. However, for all this to materialize, positive attitudes toward HIV and TB are expected. Therefore, to reduce the stigma associated with HIV and TB, it is crucial to instil the Ubuntu ethics, dignity, and respect ideals in HCPs’ mentality and work ethic. The importance of social dynamics in the fight against HIV and TB is highly acknowledged in South Africa [2]. Despite many useful findings, we identified a limitation with the population in that we might have missed the voices of level three and four students since we only included level 1 and two students in the study. As indicated previously, all of the nursing students are placed for clinical practice in the surrounding healthcare facilities and may take care of PLHIV and TB.

## 5. Conclusions

The nursing students were engaged in a workshop to understand their perceptions regarding the use of Ubuntu to fight stigma associated with HIV and TB. The findings confirmed that nursing students view Ubuntu as a critical component in the fight against HIV and TB. Our results also highlight the importance of using the strength of social constructs, which can be incorporated into the initiatives and programs that fight against HIV and TB stigma. It is also critical to remember that Ubuntu, as a social construct in South Africa, can be used to inform and de-stigmatize HIV and TB among local populations. Therefore, there is an urgent need to integrate Ubuntu values and principles when dealing with HIV and TB to improve care, especially in Africa, where the concept is positively accepted. Future prospects can also include investigating Ubuntu in different healthcare contexts or quantitative studies to supplement the qualitative findings.

## Data Availability

The data presented in this study are available on request from the corresponding author.

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
