# Peer review of "Ubuntu Is a Critical Component in the Fight against Human Immunodeficiency Virus and Tuberculosis Stigma: Nursing Students’ Perceptions"

_ijerph, 2024, doi:10.3390/ijerph21020229_

Round 1

Reviewer 1 Report

Comments and Suggestions for Authors

Overall, the manuscript “Ubuntu Is a Critical Component in the Fight against HIV and TB Stigma: Nursing Students' Perceptions” provides valuable insights into the role of Ubuntu in addressing HIV and TB stigma among nursing students. However, it would benefit from incorporating my suggestions as indicated below.

ABSTRACT:

The following grammatical issues need to be corrected:

·     The structure of lines 12-13 is incorrect. It should read “Ubuntu values and principles, we found, assist in reducing this problem in many African regions.”

·     Line 19, change “expression” to “expressions”: “Two primary themes emerged as expression(s) of Ubuntu…”

·     Line 21, change “that that” to “that”: “This confirms that within Ubuntu, there are elements that that might…”

It is recommended that the abstract includes a brief definition or explanation of “Ubuntu” to cater to an international readership. Since “Ubuntu” is a culturally specific concept, not widely known outside certain regions, providing this clarification at the outset would enhance the accessibility and comprehension of the study's context and significance for readers unfamiliar with the term. (I would recommend a short explanation after the second sentence.)

INTRODUCTION:

Line 30: The term “tuberculosis” should be abbreviated here at first mention.

Lines 30-31: The term “Mycobacterium tuberculosis” should be italicised. This sentence is also missing commas: “…an opportunistic disease known as tuberculosis, which is caused by the Mycobacterium tuberculosis complex, is affecting…”

Lines 36-37 appear to be somewhat mis-phrased or awkwardly constructed, making it challenging to understand. Do the authors perhaps mean, “Hence, [a] study that unpacked…”? The current sentence construction makes it seem like they are referring to their presented study. The next line (line 38) is also grammatically incorrect as it should be “ The same authors’ study…”, taking note of the fact that this study was not that of a single author.

Lines 41-42 has missing commas: “Another study discovered that HCPs, including nurses and nursing students, compromise…”

Line 51 mentions the “95-95-95 targets” of UNAIDS, but there is no explanation of this, leaving readers unfamiliar with this specific UNAIDS initiative uncertain about its meaning and significance. Therefore, it is suggested that the authors include a brief explanation to improve the manuscript's clarity and accessibility, making it more useful and informative for a diverse readership. Consider, “Additionally, it will be challenging to meet the UNAIDS 95-95-95 targets by 2025, which aim for 95% of people living with HIV to know their status, 95% of those diagnosed to be on antiretroviral therapy, and 95% of those on therapy to achieve viral suppression.”

Although the introduction provides a comprehensive background on the prevalence and impact of HIV and TB stigma, especially in the context of South Africa, as well as the relevance of Ubuntu in this context, it could benefit from a more concise presentation of other key points to enhance readability and focus. The introduction should more directly articulate the specific research gap your study addresses and the unique insights it offers, particularly in exploring nursing students' perceptions of Ubuntu in combating HIV and TB stigma. Additionally, the importance and potential impact of your study on healthcare practices, especially in the context of HIV and TB in South Africa, could be more explicitly highlighted. This would not only clarify the study's objectives but also its significance. As it stands, the detailed background, while informative, might dilute the focus on these critical aspects, so streamlining the content to emphasize the research gap and the study's importance would provide a clearer, more impactful introduction.

MATERIALS AND METHODS:

In the “Research Design and Setting” section (line 93), the use of qualitative participatory research is appropriately chosen for the study’s aims. However, more details on the research setting would enhance understanding. Although they mention a workshop setting in the “Procedures” section, it would be beneficial to expand on the details of the research setting in the earlier section. Providing specifics about the location and environment (geographical and cultural context), characteristics, and logistical details of where the workshops were conducted would offer readers a more comprehensive understanding of the study's context. Such information can be crucial in qualitative research, as it helps elucidate how the setting may have influenced participant responses and the overall outcomes of the study.

The “Participants and Sampling” section (line 96), provides detail on the sampling method and participant demographics. However, for better clarity on the study's context and applicability, a more explicit explanation of the participants' cultural and ethnic backgrounds would be beneficial. Given the diverse nature of South Africa, which includes 11 official languages and a wide range of cultural and racial groups, it is important to know if the participants were representative of this diversity. Were participants from various cultural, racial, or ethnic groups, including both African and non-African origins, included in the study? Clarifying this aspect would strengthen the validity of the study, particularly in understanding how the concept of Ubuntu is perceived across different segments of the South African population. Including the ethnic or racial distribution of the participants, if available, would provide valuable insights into the study's findings and their broader implications. (A graph, chart, or other visual aid might be a beneficial addition here to aid the reader.)

In lines 121-123, the two questions should be highlighted as either “1.”, or “1)”, not a combination of these as “1).”.

Lines 126-127 should be corrected, “All the points were captured both audibly (through recordings) and visually on the flipcharts.” (Also note that the style “flipcharts” [one word] is used in line 127 while the style “flip charts” [two words] is used in line 129. The authors should check the manuscript for consistency.)

In the “Data Analysis” section (line 128), the process is comprehensively outlined, providing a solid basis. To further strengthen this section, consider providing more detail of how specific data excerpts led to the formation of certain categories and themes (i.e., the transition from initial codes to final themes). Additionally, a brief explanation of how the research team addressed potential biases during the coding and theme development process would enhance the robustness of your methodology.

Addition of commas to enhance readability:

·     Line 125: “…responses to everyone, and when needed, probing questions…”

·     Line 131: “were understood, we… for coding using Excel.”

·     Line 135: “research team discussions, categories…”

·     Line 139: “…the nursing students quotes…”

In the “Trustworthiness” section (line 140), the first sentence should rather have an in-text reference.

RESULTS:

Lines 154-155, 169-171, 174-175, and elsewhere, the font should not be italic, since it is not consistent with other quotes provided in the manuscript. Alternatively, other quotes should also be italicised.

In line 167, the abbreviation “PLHIV” is used; this should be defined at first mention (in line 45).

In line 195, the phrase “Group 18 Group 17” should be corrected: One of these should be removed, based on which group said the provided statement (it seems, based on the next paragraph that the correct group here is Group 17). This statement also seems to be incorrect; surely the group did not mean that Ubuntu can prevent HIV and TB (the actual diseases), but rather that it can prevent HIV and TB stigma. Thus, the word “stigma” should be added, “Ubuntu is well positioned to prevent HIV and TB (stigma), because…”

Line 200, hyphenate the term “TB-related”. Line 200 also needs linguistic correction, “…providing the clients with [a] dignified service[s]…”. The authors would need to check the group’s actual wording.

DISCUSSION:

The discussion section effectively ties the results back to the research question and existing literature. It emphasizes the importance of Ubuntu in addressing HIV and TB stigma. However, it could be improved by discussing potential limitations and implications for practice more explicitly. This includes limitations due to the sample size, the qualitative nature of the study, or any potential biases. Acknowledging these limitations can provide a more balanced view of the research and its implications.

Lines 245-247 have missing words or grammatical issues that affect its clarity. Consider rephrasing to: A study that exported ancient African wisdom into the global world argues that it is hard to pin down the African understanding of Ubuntu in one language because it derives its knowledge from the real world in which it grows [30].”

In line 278 change “Hospital’s” to “Hospital”.

In line 280 change “PLHIV” to “PLHIV’s”.

Missing comma in line 292, “…these to materialise, positive attitudes…”

In line 294, the previously provided abbreviation (see line 39) should be used for HCP. The grammar should also be corrected to “…health care providers’ mentality…” (or HCPs’ mentality).

CONCLUSION:

The conclusion effectively summarizes the key findings and their implications, especially regarding the integration of Ubuntu in healthcare within the African context. However, it would benefit from a more detailed discussion of potential areas for future research. In this regard, the brief reference to the recent publication by Olivier and Luies (2023), “WHO Goals and Beyond: Managing HIV/TB Co-infection in South Africa” (https://doi.org/10.1007/s42399-023-01568-z), could be expanded upon. This publication, which also addresses the issues of stigma and the importance of education and destigmatization in the context of HIV and TB, offers valuable insights that align closely with your study's themes. By incorporating more aspects from Olivier and Luies' work, your manuscript could deepen its discussion on these topics. Also, considering the insights from this publication might provide a more comprehensive perspective on future directions and implications stemming from your research.

Future prospects can also, for example, include exploring Ubuntu in different healthcare contexts, or quantitative studies to supplement the qualitative findings.

GENERAL ISSUES:

The manuscript should have consistent spelling conventions throughout, as there is currently a mix of UK and US English spellings (e.g., “recognised” vs. “recognized”; “stigmatised” vs. “stigmatized”; “utilise” vs. “utilize”; “characterise” vs. characterized” and other terms). Aligning all spellings to either UK or US English, based on the journal's preference or the authors' choice, would enhance the professionalism and readability of the manuscript.

In terms of general grammar and language corrections, the manuscript should be reviewed in its entirety as some sentences are lengthy and could be broken down for clarity and ease of reading.

REASONS FOR THE EVALUATION OF MANUSCRIPT COMPONENTS AS PER JOURNAL INSTRUCTIONS

·     Introduction: Can be improved.

[Reason: While the introduction provides a comprehensive background, particularly on the prevalence and impact of HIV and TB stigma and the significance of Ubuntu, it could benefit from more concise presentation and a clearer statement of the research gap and study objectives]

·     Cited References: Yes.

  [Reason: References are relevant and appear to support the research adequately.]

·     Research Design Appropriate: Yes.

[Reason: The qualitative participatory research design is appropriate for exploring nursing students' perceptions, aligning well with the study's aims to understand subjective experiences and perspectives.]

·     Methods Description: Must be improved.

[Reason: While the methods are generally well-described, especially the data collection process through interactive workshops, more details could be provided in the research setting and the process of deriving and validating themes in data analysis.]

·     Results Presentation Clear: Yes.

[Reason: The results are clearly presented and thematically organised, with the inclusion of direct quotes from participants adding depth and authenticity to the findings.]

·     Conclusions Supported by Results: Yes.

[Reason: The conclusions drawn in the study are supported by the results. They effectively summarize the key findings and suggest practical implications, particularly the integration of Ubuntu in healthcare. However, the conclusion could be further strengthened by a more explicit discussion of potential areas for future research.]

Comments on the Quality of English Language

Language corrections is indicated in previous feedback.

Author Response

Dear reviewer,

Thank you very much for your constructive feedback. The provided comments assisted to shape this manuscript.  All the comments are accepted and addressed. See the attached document. 

Reviewer 2 Report

Comments and Suggestions for Authors

Thank you for the nice possibility to read this highly interesting paper. Topic of the article is very important and relevant. Methods are innovative and I think that data is well analyzed. However, I would like to see larger results. Chapter of discussion is very good but previous chapter of results is not on same level. I ask the authors enlarge the results because it would be very interesting to read more. Additionally, I hope the authors could write more in the introduction, or even add one chapter more about theoretical background. The introduction is informative but it is short. Ubuntu is totally new concept for me, while I´m not African. The concept is very interesting and I thought that perhaps it would be good to write an own chapter about this concept. 

All in all, it was a great pleasure to read this article. I hope the authors are able to enlarge the text. After it I am very happy to accept this article.

Author Response

Dear reviewer,

Thank you for reviewing our manuscript. The provided comments are appreciated and were used to improve the manuscript. See the attached document. 
